# Uncertainty-Aware Pseudo-Labeling and Dual Graph Driven Network for Incomplete Multi-View Multi-Label Classification

## Abstract

Multi-view multi-label classification has recently received extensive attention due to its wide-ranging applications across various fields, such as medical imaging and bioinformatics. However, views and labels are usually incomplete in practical scenarios, attributed to the uncertainties in data collection and manual labeling. To cope with this issue, we propose an uncertainty-aware pseudo-labeling and dual graph driven network (UPDGD-Net), which can fully leverage the supervised information of the available labels and feature information of available views. Different from the existing works, we leverage the label matrix to impose dual graph constraints on the embedded features of both view-level and label-level, which enables the method to maintain the inherent structure of the real data during the feature extraction stage. Furthermore, our network incorporates an uncertainty-aware pseudo-labeling strategy to fill the missing labels, which not only addresses the learning issue of incomplete multi-labels but also enables the method to explore more reliable supervised information to guide the network training. Extensive experiments on five datasets demonstrate that our method outperforms other state-of-the-art methods.

## CCS Concepts

• **Computing methodologies** → *Supervised learning by classification.*

## Keywords

incomplete multi-view learning, incomplete multi-label classification, graph constraint, pseudo-labeling

## 1 Introduction

With the development of data acquisition technology, many researchers have found that multi-view data integrated from different sources can provide a more nuanced and diverse representation of an object. For example, an image can be described by different feature descriptors such as SIFT, Gist, and HSV. As a result, multi-view learning has emerged as an important approach in the realm of data analysis, and many related works are based on subspace learning [21] [23] [26] and matrix factorization [13] [25] have been proposed. On the other hand, as a classical classification problem, multi-label classification (MLC) has also gaining prominence in the field of pattern recognition for a long time. Unlike single-label data

**Unpublished working draft. Not for distribution.**

which requires mutual exclusivity among labels, multi-label data contains various category tags and naturally maintains complex label correlations. For instance, an image might be described by multiple elements such as "neon signs", "sidewalks", and "passing cars", each contributing to a more comprehensive understanding of the scene. However, traditional multi-label classification methods often suffer from a significant issue as they primarily depend on features extracted from a single perspective, which limits their performance. By integrating multi-view learning into multi-label classification, this limitation can be effectively mitigated. As a result, a composite multi-view multi-label classification (MvMLC) has emerged and attracted increasing attention from researchers.

MVMLC synergistically combines the strengths of both multi-view learning and multi-label classification, offering a more robust and comprehensive framework for understanding complex multi-view multi-label data and plenty of related works have been proposed in recent years. For example, Sun and Zong proposed a Latent Conditional Bernoulli Mixture model called LCBM, which leverages a latent conditional Bernoulli mixture approach to integrate features from multiple views and using a shared latent subspace for label dependency construction [18]. Zhang et al. proposed a Latent Semantic Aware Multi-view Multi-label Learning approach, which utilizes matrix factorization and Hilbert-Schmidt Independence Criterion in kernel spaces to align latent semantic components across multiple views [27]. However, a significant challenge in MvMLC is the frequent occurrence of incomplete data, both in terms of views and labels. This challenge directly leads to the necessity of Incomplete Multi-View Multi-Label Classification (iMvMLC), which focuses on scenarios where some views or labels are missing. For instance, Tan et al. proposed the iMVWL, which attempts to learn a shared subspace that integrates weak label information and local label correlations [19]. Li et al. proposed an iMvMLC method based on matrix factorization, named NAIM3L, which bridges non-aligned views through common labels and exploits both global and local structural relations within multiple labels [9]. More recently, DNN-based methods have been introduced, offering improved performance by leveraging high-level feature extraction and complex model architectures, such as LMVCAT [12] and DICNet [11]. Despite these advances, existing methods struggle to fully utilize supervision information to help the model extract more discriminative features due to the incompleteness problem.

To bridge this gap, our research introduces a novel network called UPDGD-Net for the iMvMLC task. Different from the existing works, the proposed UPDGD-Net tries to impute missing labels with credible pseudo-labels through an uncertainty-aware pseudo-labeling strategy and then utilizes the filled label matrix to impose dual graph constraints on high-dimensional features extracted by the model. Specifically, we first design two distinct transformer-based modules, one dedicated to cross-view aggregation and the other for multi-label classification. Additionally, we apply average

view token (AVT) in the former to better learn the consistent information across multiple views. Furthermore, our network harnesses the semantic guidance offered by the label matrix to largely maintain the intrinsic relationship of the original data by constraining the embedded features of both view-level and label-level. Last but not least, inspired by the self-paced learning [14] and uncertainty analysis from the Monte-Carlo Dropout [3], we propose a novel uncertainty-aware pseudo-labeling strategy that maximizes the use of available data while maintaining the integrity and richness of the multi-label information. Overall, compared to existing methods, the contributions of our method can be summarized as follows:

- To address the challenge of incomplete data learning, we propose the UPGDG-Net that incorporates information contained within missing labels through a novel uncertainty-aware pseudo-labeling strategy. UPDGD-Net attempts to maximize the use of available data while maintaining the integrity and richness of the multi-label information, thus enhancing the accuracy of the classification.
- We introduce a label-guided dual-constraints approach in our network, which capitalizes on the untapped potential of multi-label semantics to preserve the intrinsic relationship of the original data in the embedding space.
- We apply average view token (AVT) in our transformers-based architecture to better learn and synthesize the consistent information across multiple views. Extensive experiment results on five datasets confirm the effectiveness of our method.

## 2 Related work

### 2.1 Multi-View Multi-Label Classification

Multi-view Multi-label classification(MvMLC) is the combination of multi-view learning and multi-label classification which need to handle multi-view multi-label data at the same time and thus increase the complexity of this issue. Compared to traditional single-view multi-label classification, applying multi-view learning to MLC tasks has demonstrated superior performance, and thus many related works have been proposed. For example, Zhao et al. introduced LVSL, enhancing consistency and diversity in multi-view learning by leveraging both global and local structural label information [28]. Zhao et al. developed CDMM, an approach integrating the Hilbert–Schmidt Independence Criterion to incorporate label correlation and view contribution factors [29]. Liu et al. developed a multi-view multi-label learning method named ELSMML that enhances label correlation understanding through a high-order strategy label correlation matrix and integrates multi-view learning with dimension reduction for capturing high-level semantic label and latent feature information. [10] Tan et al. proposed a deep learning method for MvMLC that exploits both shared subspace and view-specific information to integrate diverse representations and semantics [20]. Ma et al. proposed MMC-GFLS that addresses the limitations of global feature and label selection by incorporating local patterns through group-specific feature selection and label correlations.[16] Zhang et al. introduced LSA-MML, a matrix factorization technique to align the latent semantic components across views, effectively managing multi-view data complexity and preserving label integrity [27].

## 2.2 Incomplete Multi-view Multi-label Classification

Previous studies in multi-view multi-label classification (MvMLC) have primarily relied on the completeness assumption of multi-view multi-label data, which neglects the incompleteness scenarios in real-world applications. As a result, the incomplete multi-view multi-label classification (iMvMLC) task has received attention from plenty of researchers and corresponding methods need to consider the incompleteness issue in multi-view multi-label data. For instance, Tan et al. proposed iMvML, which learns a shared subspace that includes weak label information and local label correlations to manage cross-view relationships and incomplete label scenarios effectively [19]. Li and Chen introduced NAIM3L, focusing on both the global high rankness and the local low rankness in the label embedding space [9]. Zhu et al. proposed a method named WCC-MVML-ID, which integrates within-view, cross-view, and consensus-view representations to effectively process incomplete multi-view multi-label datasets. [30]. In addition to these traditional methods, DNN-based frameworks in iMvMLC have also shown significant performance. For example, Liu et al. proposed the deep iMvPMLC framework DICNet, which utilizes stacked autoencoders for view-specific feature extraction and introduces an incomplete instance-level contrastive learning scheme [11]. LMVCAT leverages the self-attention mechanism to extract high-level features and exploit inter-class correlations to enhance classification performance [12].

## 2.3 Notations and Problem Definition

Assume that we are given a multi-view data $\left\{X^{(v)} \in \mathbb{R}^{n \times d_v}\right\}_{v=1}^{m}$, in which $m$ denotes the number of views and $d_v$ represents the original feature dimension of the $v$-th view. And we define $Y \in \{0, 1\}^{n \times c}$ as label matrix, where $c$ denotes the number of categories. Specifically, $Y_{i,j} = 1$ represents the $i$-th sample is annotated as $j$-th category, otherwise $Y_{i,j} = 0$. Furthermore, we design two prior matrices, $W \in \{0, 1\}^{n \times m}$ denotes the missing-view indicator and $G \in \{0, 1\}^{n \times c}$ denotes the missing-label indicator, where $W_{i,j} = 1$ and $G_{i,j} = 1$ represent the $j$-th view and category of the $i$-th sample is available, respectively, and $W_{i,j} = 0$ and $G_{i,j} = 0$ mean the $j$-th view and category of the $i$-th sample is missing, respectively. To simplify, the missing data in $\left\{X^{(v)} \in \mathbb{R}^{n \times d_v}\right\}_{v=1}^{m}$ and $Y \in \{0, 1\}^{n \times c}$ is initialized as '0' in the data-preparation stage. The target of our method is to train a multi-label classification network on incomplete multi-view partial multi-label data, which can accurately predict the categories of unlabeled incomplete multi-view data.

## 3 The Proposed Method

In this section, we will introduce each component of the proposed UPDGD-Net, including a multi-view semantic representation learning framework, dynamically weighted fusion module, multi-label interaction module, label-guided dual constraints, and uncertainty-aware pseudo-labeling module. Fig.1 shows the framework of our UPDGD-Net and Fig.2 shows the process of our uncertainty-aware pseudo-labeling strategy.

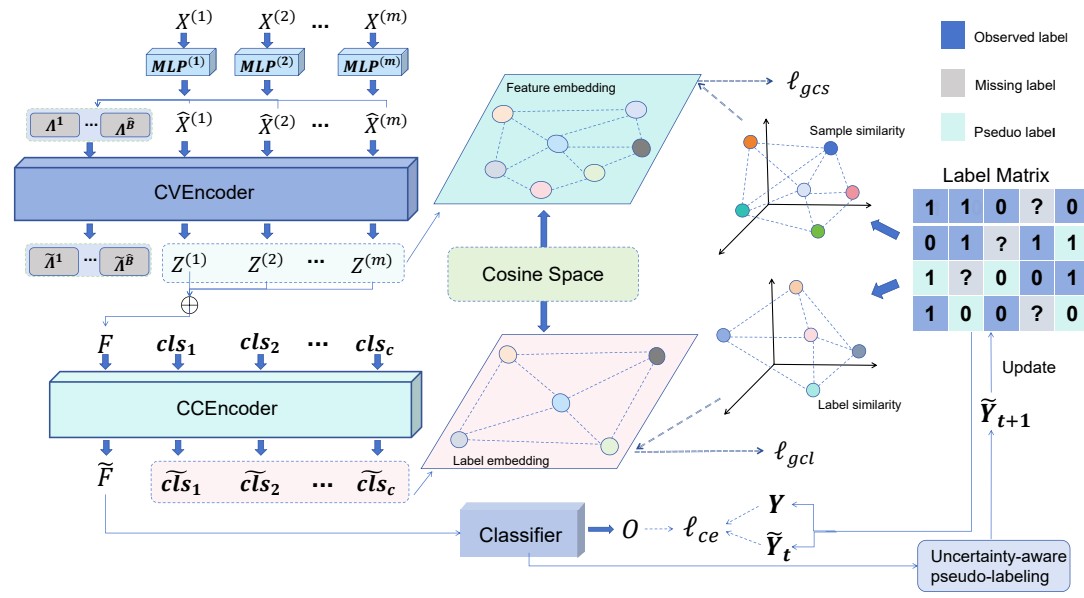

**Figure 1: An overview of our UPDGD-Net. In our network, CVEncoder and CCEncoder both leverage transformer-based architectures to facilitate cross-view and cross-label interaction respectively. Label-guided dual constraints exploit semantic information within multi-label to construct graph constraints on embedding features and the detail of our uncertainty-aware pseudo-labeling strategy is shown in Figure 2**

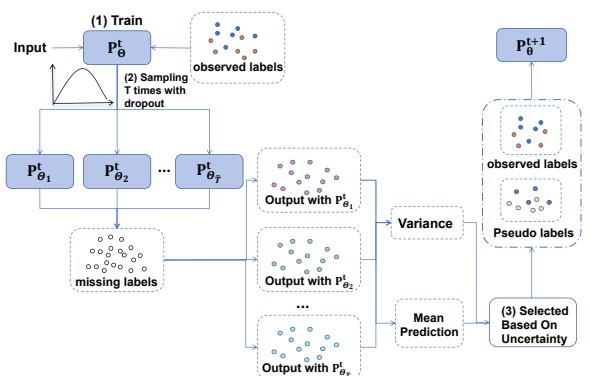

**Figure 2: An overview of our pseudo-labeling strategy. Given trained model parameters $P_\Theta^t$ during $t$-th epoch, we can sample Bernoulli distribution $\hat{T}$ times using the dropout technique and obtain $\hat{T}$ groups of model parameter sets. Then we perform stochastic forward propagation $\hat{T}$ times with these parameter sets and obtain $\hat{T}$ sets of output. Then we calculate the mean values and variance of outputs. Finally, we calculate the uncertainty of outputs based on variance and select pseudo-labels for $\hat{T} + 1$-th epoch from mean values based on uncertainty.**

## 3.1 Multi-View Semantic Representation Learning Framework

As is well-known, the success of multi-view learning lies in exploiting the consistency and complementary information of multi-view

data. To achieve this, similar to [12], we design a new cross-view transformer encoder (CVEncoder) to facilitate the cross-view interaction and extract high-quality cross-view features. To ensure a uniform feature dimension across various views, we first employ Multilayer Perceptrons (MLP) to map the original multi-view data into a shared embedding space denoted as $\{\Phi^{(v)} : X^{(v)} \in \mathbb{R}^{n \times d_v} \to \hat{X}^{(v)} \in \mathbb{R}^{n \times d_e}\}_{v=1}^m$, where $d_e$ denotes the mapped dimension of multi-view data. Additionally, to ensure the consistency of multi-view features, we introduce some identical Average View Tokens (AVT) calculated by the average of all available views of each sample as extra inputs for CVEncoder. The calculation of the AVT corresponding to the $i$-th sample is formulated as follows:

$$\Lambda_i = \frac{1}{\sum_{v=1}^m W_{i,v}} \sum_{v=1}^m \hat{X}_{i,:}^{(v)} W_{i,v} \qquad (1)$$

where $\Lambda_i \in \mathbb{R}^{1 \times d_e}$ represents the AVT for the $i$-th sample. In (1), the missing view indicator matrix $W \in \mathbb{R}^{n \times m}$ is introduced to disregard the missing views in each sample. Since the self-attention mechanism dynamically adjusts token influence based on the distribution of attention scores, allowing multiple identical tokens can amplify their impact within the sequence. In our method, we adopt $\hat{B}$ AVTs with the same value for each sample. Considering these tokens, the input of CVEncoder for the $i$-th sample can be expressed as: $V_i = \text{Concat}\left(\hat{X}_i; \Lambda_i^1; ...; \Lambda_i^{\hat{B}}\right)$ where $V_i \in \mathbb{R}^{(m+\hat{B}) \times d_e}$ and $\Lambda_i^1 = ... = \Lambda_i^{\hat{B}}$. Specifically, to mitigate the negative impact of missing views when calculating attention scores in the CVEncoder, we also utilize the missing view indicator matrix $W$ to mask the missing views during the computation of multi-head self-attention scores. The process for our masked multi-head self-attention layer

is detailed as follows: As shown in Fig.1, each input is linearly projected to obtain queries, keys, and values, utilizing $h$ groups of projective matrices $\left\{ P^{q_t}, P^{k_t}, P^{v_t}, \right\}_{t=1}^{h}$, where $h$ represents the number of attention heads. To effectively mask the embedding features by the missing-view distribution, we expand the mask matrix $W$ to $\tilde{W} \in \mathbb{R}^{n \times (m+\hat{B})}$ by adding $\hat{B}$ column vectors with all one. Then we construct a mask matrix of sample $i$: $M_i = \tilde{w}_i^T \tilde{w}_i \in \mathbb{R}^{(m+\hat{B}) \times (m+\hat{B})}$, where $\tilde{w}_i$ is $i$-th row vector of $\tilde{W}$. Then for $i$-th sample's input $V_i$, the view correlations $A_i^t$ and output $H_i^t$ is calculated as follows:

$$A_i^t = softmax\left( \frac{fill\left( (V_i P^{q_t})(V_i P^{k_t})^T, M_i \right)}{\sqrt{d_h}} \right) \quad (2)$$

$$H_i^t = A_i^t \left( V_i P^{v_t} \right) \quad (3)$$

where $d_h = d_e/h$ represents the dimensionality per attention head and $H_i^t \in \mathbb{R}^{(m+\hat{B}) \times d_h}$ is the output of $t$-th attention head. $P^{q_t} \in \mathbb{R}^{d_e \times d_h}$, $P^{k_t} \in \mathbb{R}^{d_e \times d_h}$, and $P^{v_t} \in \mathbb{R}^{d_e \times d_h}$ are projective matrices. Additionally, we introduce a *fill* function $fill(A, B)$ before the softmax active function in the CVEncoder to set $A_{i,j} = -1e^9$ if $B_{i,j} = 0$. With such a *fill* operation, the softmax can ignore the negative affect of missing views when calculating the attention scores. Finally, we obtain the output of CVEncoder for sample $i$ by concatenating the outputs of $h$ attention heads: $\tilde{V}_i = Concat(H_i^1, ..., H_i^h) \in \mathbb{R}^{(m+B) \times d_e}$. Overall, our CVEncoder can be simply formulated as: $\Gamma : \left\{ V_i \in \mathbb{R}^{(m+\hat{B}) \times d_e} \right\}_{i=1}^{n} \rightarrow \left\{ \tilde{V}_i \in \mathbb{R}^{(m+\hat{B}) \times d_e} \right\}_{i=1}^{n}$ and the first $m$ rows of CVEncoder's output $\left\{ \tilde{V}_i \in \mathbb{R}^{(m+\hat{B}) \times d_e} \right\}_{i=1}^{n}$, i.e., $\{Z_i \in \mathbb{R}^{m \times d_e}\}_{i=1}^{n}$ is regarded as the embedding feature of each sample.

## 3.2 Dynamically Weighted Fusion Module and Multi-label Interaction Module

After obtaining the embedding features of each view using CVEncoder, our goal is to create a unified and consistent representation that fully characterizes the multi-view data. However, different views may contribute differently to this representation. To account for varying discriminative abilities among multiple views, we adopt a dynamically weighted fusion approach inspired by [12]. We define the fused feature $f_i$ for sample $i$ as follows:

$$f_i = \sum_{v=1}^{m} \frac{e^{a_v} z_i^{(v)} W_{i,v}}{\sum_v e^{a_v} W_{i,v}} \quad (4)$$

where $z_i^{(v)}$ denotes the embedding feature of the $v$-th view/row in $Z_i$ and $a_v$ represents a learnable scalar weight for the $v$-th view and $a_v$ is updated by network. Notably, to mitigate the negative impact of missing views during multi-view fusion, we introduce a missing-view indicator matrix $W$ into our fusion module. This equation captures an adaptive weighting mechanism for each view, resulting in a comprehensive feature representation that incorporates information from all available views.

When we obtain a consistent common representation from incomplete multi-view data, another crucial issue is how to utilize the relationships between multiple labels to enhance the discriminative power of the multi-view representation. To achieve this purpose,

inspired by [7]. we introduce label embeddings to map each category directly into the feature space, leveraging the self-attention mechanism to capture category correlations. More specifically, we randomly initialize $c$ label embedding features as $c$ class tokens $\left\{ cls_i \in \mathbb{R}^{d_e} \right\}_{i=1}^{c}$, and introduce them into a cross-category transformer encoder (CCEncoder) along with the fusion features of samples. The input to the CCEncoder is thus $\left\{ C_i \in \mathbb{R}^{(c+1) \times d_e} \right\}_{i=1}^{n}$ where $C_i = Concat\left(f_i; cls_1; ...; cls_c\right)$. The CCEncoder, like CVEncoder, fosters information exchange between fusion features and class tokens. This process ensures that view-fusion features capture subcategory information based on similarities, aligning them closely with class tokens. The CCEncoder's output is $\left\{ \tilde{C}_i \in \mathbb{R}^{(c+1) \times d_e} \right\}_{i=1}^{n}$ and the first row in $\left\{ \tilde{C}_i \right\}_{i=1}^{n}$, *i.e.*, $\left\{ \tilde{f}_i \in \mathbb{R}^{1 \times d_e} \right\}_{i=1}^{n}$ is regarded as the final discriminative feature for classification. Finally, our CCEncoder can be simply formulated as: $\Upsilon : \left\{ C_i \in \mathbb{R}^{(c+1) \times d_e} \right\}_{i=1}^{n} \rightarrow \left\{ \tilde{C}_i \in \mathbb{R}^{(c+1) \times d_e} \right\}_{i=1}^{n}$

## 3.3 Label-Guided Dual Constraints

In our work, we attempt to introduce two label-guided constraints to enhance the quality of the latent representation and the initialized class tokens.

**Label-guided constraint on latent representations**: Generally speaking, original data typically exhibit a natural topological structure, where a part of samples (such as nearest neighbor samples) or categories share similarities while differing from others. For example, some samples may simultaneously annotated with two or more identical labels. Unfortunately, most deep learning methods overlook this structure when transforming data into feature space. To address the above issue and obtain a more reasonable representation yet good classification performance, following [12], we introduce a label-guided constraint on latent representations $\left\{ Z_i \in \mathbb{R}^{m \times d_e} \right\}_{i=1}^{n}$ that leverages the rich semantic information of multiple categories to guide the network training. Firstly, based on the smoothness assumption that if two samples have more identical labels, they should also have more similarity [24], we construct a label-based correlation graph $Q \in \mathbb{R}^{n \times n}$ according to the available label data as follows:

$$Q = (Y \odot G)(Y \odot G)^T ./ GG^T \quad (5)$$

where $\odot$ denotes Hadamard product. $GG^T$ is used for normalization in the calculation of the label-based correlation matrix $Q$. This equation quantifies the similarity between samples based on the number of common positive labels they share. In simpler terms, two samples are considered more similar if they share a larger number of common positive tags. To this end, we further calculate the similarity between two embedding features within each view using cosine similarity:

$$S_{i,j}^{(v)} = \frac{z_i^{(v)} z_j^{(v)T} + 1}{2 \left\| z_i^{(v)} \right\| \left\| z_j^{(v)} \right\|} \quad (6)$$

where $z_i^{(v)}$ and $z_j^{(v)}$ are the embedding features of the $v$-th view of the $i$-th and $j$-th sample that are output by CVEncoder. Then

we introduce the following graph-based cross-entropy loss to align the structure of latent representation to the target structure of supervised label-based graph $Q$:

$$\ell_{gcs} = -\frac{1}{2\sum_{v=1}^{m}\sum_{\substack{i=1\\j\neq i}}^{n} W_{i,v}W_{j,v}} \sum_{v=1}^{m}\sum_{i=1}^{n}\sum_{j\neq i}^{n} \left(Q_{i,j}\right.$$

$$\left. \log S_{i,j}^{(v)} + (1 - Q_{i,j})\log(1 - S_{i,j}^{(v)})\right)(W_{i,v}W_{j,v}) \quad (7)$$

where $\sum_{v=1}^{m}\sum_{i=1,j\neq i}^{n} W_{i,v}W_{j,v}$ represents the count of available sample pairs across all $m$ views.

**Label-guided constraint on class tokens**: In our network, we introduce $c$ class tokens as input of the CCencoder to improve the interaction of the multi-view features and labels. Correspondingly, we obtain $c$ label embedding features, which can be regarded as the discriminative representation of $c$ categories. For these label embeddings, we also expect them to preserve similar label correlations as the original labels. To this end, similar to [5], we introduce a label-guided constraint on these label embeddings. Firstly, we construct a new label correlation graph $\mathfrak{I} \in \mathbb{R}^{c \times c}$ in which each element represents the probability of a sample having both the $i$-th and $j$-th labels simultaneously as follows:

$$\mathfrak{I}_{i,j} = \frac{\sum_{k=1}^{n} Y_{k,i}Y_{k,j}}{2\sum_{k=1}^{n}(Y_{k,i} + Y_{k,j})} \quad (8)$$

where $\sum_{k=1}^{n} Y_{k,i}Y_{k,j}$ calculates the total number of co-occurrences of category $i$ and category $j$. $\sum_{k=1}^{n}(Y_{k,i} + Y_{k,j})$ refer to the sum of the number of occurrences of category $i$ and category $j$.

Similarly, for the label embedding features corresponding to the outputted class tokens from CCEncoder, we can calculate their cosine distance-based similarities as follows:

$$K_{i,j} = \frac{\tilde{cls}_i \tilde{cls}_j^{T} + 1}{2\left\|\tilde{cls}_i\right\|\left\|\tilde{cls}_j\right\|} \quad (9)$$

where $\tilde{cls}_i$ denotes the label embedding of $i$-th category.

Then, to preserve the similar structure as labels, we introduce the following loss on the label embeddings:

$$\ell_{gcl} = \frac{1}{c(c-1)} \sum_{i=1}^{c}\sum_{j\neq i}^{c} \left(K_{i,j} - \mathfrak{I}_{i,j}\right)^2 \quad (10)$$

## 3.4 Uncertainty-Aware Pseudo-Labeling and Classifier

Pseudo-label strategy has proven its effectiveness in semi-supervised learning and unsupervised learning tasks. For the incomplete label learning network, we also design an uncertainty-aware pseudo-labeling strategy to explore more supervised information of the incomplete label data. In the designation of our pseudo-labeling strategy, the key challenges are evaluating the uncertainty of the predicted labels and selecting the credible pseudo-label to guide the model training. Inspired by [3], we apply the Monte-Carlo Dropout (MC dropout) approach to quantify the uncertainty of the predicted labels. Dropout is implemented by probabilistically deactivating neurons using a Bernoulli distribution, making the network's output $y^*$ for an input $x^*$ follow a predictive distribution rather than a definite value. Considering a deep neural network with $L$ layers, each layer has $N_i$ neurons in the $i$-th layer, the network parameters

is denoted as $\{\theta_i\}_{i=1}^{L}$ where $\theta_i = \{\theta_{i,j}\}_{j=1}^{N_i}$. According to standard dropout, which follows a Bernoulli distribution, we approximate the prediction distribution by sampling this distribution $\hat{T}$ times and obtain $\hat{T}$ sets of binary vectors, each representing a unique dropout operation across the layers, denoted as $\left\{\vartheta_1^t, \vartheta_2^t, ...\vartheta_L^t\right\}_{t=1}^{\hat{T}}$ where $\vartheta_i^t = \left\{\vartheta_{ij}^t\right\}_{j=1}^{N_i}$ is the binary vector for the $i$-th layer in the $t$-th dropout operation. Each neuron's dropout, $\vartheta_{i,j}^t$, is drawn from a Bernoulli distribution $\vartheta_{i,j}^t \sim Bernoulli(p)$, with $p$ equal to the dropout ratio used during training. Then we can obtain $\hat{T}$ groups of network parameter sets as follows:

$$\left\{\Theta_1^t, \Theta_2^t, ...\Theta_L^t\right\}_{t=1}^{\hat{T}} = \left\{\theta_1 \circ \vartheta_1^t, \theta_2 \circ \vartheta_2^t, ..., \theta_L \circ \vartheta_L^t\right\}_{t=1}^{\hat{T}} \quad (11)$$

where $\circ$ denotes the Hadamard product.

As shown in Fig.2, given dataset $X$ with $n$ samples, we can perform stochastic forward propagation $T$ times with $\left\{\Theta_1^t, \Theta_2^t, ...\Theta_L^t\right\}_{t=1}^{\hat{T}}$ and obtain $\hat{T}$ sets of outputs $\left\{\hat{Y}^1, ..., \hat{Y}^{\hat{T}}\right\}$, where $\hat{Y}^{\hat{t}} \in \mathbb{R}^{n \times c}$. We can estimate the mean of the outputs as follows:

$$\mathbb{E}_{q\left(\hat{Y}|X\right)}\left(\hat{Y}_{i,j}\right) \approx \frac{1}{\hat{T}}\sum_{t=1}^{\hat{T}} \hat{Y}_{i,j}^t \quad (12)$$

where $q\left(\hat{Y}|X\right)$ represents the approximate predictive distribution after $\hat{T}$ sampling times. Then we estimate:

$$\mathbb{E}_{q\left(\hat{Y}|X\right)}\left(\left(\hat{Y}_{ij}\right)^2\right) \approx \frac{1}{\hat{T}}\sum_{t=1}^{T} \left(\hat{Y}_{i,j}^t\right)^2 \quad (13)$$

Finally, we can calculate the variance of the model's output:

$$Var_{q\left(\hat{Y}|X\right)}(\hat{Y}_{i,j}) \approx \mathbb{E}_{q\left(\hat{Y}|X\right)}\left(\left(\hat{Y}_{i,j}\right)^2\right)$$
$$- \left(\mathbb{E}_{q\left(\hat{Y}|X\right)}\left(\hat{Y}_{i,j}\right)\right)^2 \quad (14)$$

Then we can normalize the variance to obtain the uncertainty of the output:

$$U_{i,j} = \frac{Var_{q\left(\hat{Y}|X\right)}(\hat{Y}_{i,j}) - \left(Var_{q\left(\hat{Y}|X\right)}\hat{Y}\right)_{min}}{\left(Var_{q\left(\hat{Y}|X\right)}\hat{Y}\right)_{max} - \left(Var_{q\left(\hat{Y}|X\right)}\hat{Y}\right)_{min}} \quad (15)$$

where $U_{i,j}$ denotes the uncertainty of the $j$-th output of the $i$-th sample and $U \in [0,1]^{n \times c}$. After that, we can treat the predictive mean of $\hat{T}$ outputs $\mathbb{E}_{q\left(\hat{Y}|X\right)}\left(\hat{Y}\right)$ as candidate pseudo-labels and select credible pseudo-labels from it based on uncertainty. Inspired by [15], we apply a self-paced selection strategy to select these reliable pseudo-labels. Firstly, we introduce an age parameter $\lambda$ and initialize its value as 0 and progressively increment its value as follows:

$$\lambda = \frac{\log\left(1 + \tau \cdot e\right)}{\log\left(1 + \tau \cdot \hat{e}\right)} \quad (16)$$

where $e$ denotes the $e$-th epoch and $\hat{e}$ denotes the overall training epochs. Specifically, we introduce a temperature coefficient $\tau$ to control the growth rate of $\lambda$. Gradually increasing $\lambda$ allows for the

selection of more credible candidate pseudo-labels over time. Then we construct weight matrix $D \in \mathbb{R}^{n \times c}$ as follows:

$$D_{i,j} = \begin{cases} 1, & \text{if } U_{i,j} < \lambda, \\ 0, & \text{otherwise.} \end{cases} \quad (17)$$

where $\lambda$ is a gradually increasing dynamic threshold and we mark output positions with uncertainty less than this threshold as 1 on $D$ for subsequent selection. To select pseudo-labels for the missing label, we need to adjust weight matrix $D$ based on missing-label indicator $G$: $\tilde{D} = D \odot \tilde{G}$ where $\tilde{G}_{i,j} = 1 - G_{i,j}$ indicates the position of missing labels in label matrix. Then we can select soft pseudo-labels from candidate pseudo-labels for the next epoch:

$$\tilde{Y}_{t+1} = \mathbb{E}_{q(\hat{Y}|X)}\left(\hat{Y}\right) \odot \tilde{D} \quad (18)$$

Finally, we can obtain the final output $O = \Psi\left(\tilde{f}\right) \in [0,1]^{n \times c}$, where $\Psi$ is the final classifier. Then we compute the following multi-label classification loss using a masked binary cross-entropy function, which incorporates both observed labels and pseudo-labels as follows:

$$\begin{aligned}
\ell_{ce} = &-\frac{1}{\sum_{i,j} G_{i,j}} \Bigg( \sum_{i=1}^{n} \sum_{j=1}^{c} Y_{i,j} \log(O_{i,j}) \\
&+ (1 - Y_{i,j}) \log(1 - O_{i,j}) \Bigg) G_{i,j} - \frac{1}{\sum_{i,j} \tilde{D}_{i,j}} \\
&\left( \sum_{i=1}^{n} \sum_{j=1}^{c} \tilde{Y}_{i,j} \log(O_{i,j}) + (1 - \tilde{Y}_{i,j}) \log(1 - O_{i,j}) \right) \tilde{D}_{i,j}
\end{aligned} \quad (19)$$

where $G$ is introduced to calculate the loss from observed labels in the first term and $\tilde{Y}$ in the second term represents the soft pseudo-labels selected from the previous training round.

Overall, our total function will be:

$$\ell_{total} = \ell_{ce} + \alpha \ell_{gcs} + \beta \ell_{gcl} \quad (20)$$

where $\alpha$ and $\beta$ are penalty coefficients. The training process is described in Algorithm 1

## 4 EXPERIMENT

### 4.1 Experimental Setup

***Dataset Description and Preparation***: Following [9, 11, 19], we train and validate our model on five well-known multi-view multi-label datasets: Corel5k [1], Pascal07 [2], ESPGame [22], IAPRTC12 [4], and MIRFLICKR [6]. Each dataset includes six views represented by GIST, HSV, Hue, Sift, RGB, and LAB features extracted from their respective datasets. To simulate real-world scenarios with missing views and partial labels, we construct incomplete multi-view partial multi-label data as follows: (1) We randomly mask 50% of samples for each view, ensuring at least one available view per sample. (2) For each category, 50% of both positive and negative tags are randomly removed. Subsequently, 70% of these samples are randomly chosen as the training set.

***Method Comparison***: We evaluated our method's effectiveness by comparing it with six well-known methods: iMvWL [19], NAIM3L [9], CDMM [29], DeepIMV [8], DICNet [11], and LMVCAT [12]. iMvWL and NAIM3L are non-DNN iMvMLC models, while the rest are DNN-based. Notably, CDMM and DeepIMV, which are

---

**Algorithm 1** Training process of **UPDGD-Net**

---

**Input:** Incomplete multi-view data $\{X^v\}_{v=1}^{m}$, Incomplete multi-label matrix $Y$, missing view indicator $W$, missing label indicator $G$. Hyperparameters $\alpha$, $\beta$, $\hat{B}$, $\hat{T}$, learning rate, and training epochs $\hat{e}$.

**Initialization:** Initialize MLPs $\left\{\Phi^{(v)}\right\}_{v=1}^{m}$, CVEncoder $\Gamma$, CCEncoder $\Upsilon$, cls tokens $\{cls_1, ...cls_c\}$, classifier $\Psi$.

**for** $e$=0 **to** $\hat{e}$ **do**
  **for** v=1 **to** m **do**
    Compute $\tilde{X}^{(v)} = \Phi^{(v)}\left(X^{(v)}\right)$
  **end for**
  Compute AVT using Eq.(1).
  Compute $\tilde{V} = \Gamma(V)$.
  Extract embedding feature $Z$ from $\tilde{I}$.
  Compute $\ell_{gcs}$ using Eq.(7).
  Compute $\tilde{f}$ using Eq.(4).
  Compute $\tilde{C} = \Upsilon(C)$.
  Compute $\ell_{gcl}$ using Eq.(10).
  Extract discriminative feature $\tilde{f}$ from $\tilde{H}$.
  Compute $O = \Psi(\tilde{f})$.
  Compute $U$ using Eq.(15).
  Obtain pseudo-labels $\tilde{Y}$ by Eq.(17).
  Compute $\ell_{ce}$ using Eq.(18).
  Compute $\ell_{total}$ using Eq.(19).
  Update $\left\{\Phi^{(v)}\right\}_{v=1}^{m}$, $\Gamma$, $\Upsilon$, $\{cls_1, ...cls_c\}$, $\Psi$.
**end for**
**Output:** Trained model parameters

---

MvMLC methods, cannot handle missing data natively. To implement the two methods on the incomplete multi-view multi-label classification tasks, we used mean imputation for missing views and set '0' for unknown tags. DICNet is a contrastive learning framework with InfoNCE loss, and LMVCAT is a transformer-based method. The above two methods can handle incomplete multi-view and partial multi-label data. Parameters for these methods were set as per recommendations in their original publications for fairness.

### 4.2 Experimental Results and Analysis

In this section, we evaluate the performance of our method for the classification task by comparing it with six state-of-the-art algorithms on the five datasets. Table 1 shows the experiment results of the five evaluation metrics, in which the missing-view rate and missing-label rate are both arranged as 50%. Fig 3 shows more results related to different missing views and missing label ratios on the Corel5k dataset to investigate the impact of missing data. According to the statistical results in Table 1, it is evident to make the following observations:

- On all metrics of both five datasets, our method achieves superior performance compared with other methods, which fully demonstrates the effectiveness of our method on the iMvMLC task.
- Compared to traditional methods, the methods that are specifically designed for double incompleteness show obvious advantages compared to other methods. This indicates the

Table 1: The performance of different methods on various datasets.

| DATA | METRIC | iMvWL | NAIM3L | CDMM | DeepIMV | DICNet | LMVCAT | OURS |
|------|--------|-------|--------|------|---------|--------|--------|------|
| Corel5k | AP | 0.283±0.011 | 0.309±0.004 | 0.309±0.004 | 0.354±0.004 | 0.381±0.004 | 0.382±0.004 | **0.413±0.004** |
| | 1-RL | 0.865±0.005 | 0.878±0.002 | 0.884±0.003 | 0.863±0.005 | 0.882±0.004 | 0.880±0.002 | **0.903±0.003** |
| | AUC | 0.868±0.005 | 0.881±0.002 | 0.888±0.003 | 0.866±0.005 | 0.884±0.004 | 0.883±0.002 | **0.905±0.004** |
| | OE | 0.689±0.015 | 0.650±0.009 | 0.590±0.007 | 0.539±0.015 | 0.532±0.007 | 0.547±0.006 | **0.520±0.020** |
| | Cov | 0.298±0.008 | 0.275±0.005 | 0.277±0.007 | 0.298±0.010 | 0.273±0.011 | 0.273±0.006 | **0.223±0.008** |
| Pascal07 | AP | 0.437±0.018 | 0.488±0.003 | 0.508±0.005 | 0.548±0.008 | 0.505±0.012 | 0.519±0.005 | **0.552±0.003** |
| | 1-RL | 0.736±0.015 | 0.783±0.001 | 0.812±0.004 | 0.815±0.008 | 0.783±0.008 | 0.811±0.004 | **0.832±0.007** |
| | AUC | 0.767±0.015 | 0.811±0.001 | 0.838±0.003 | 0.835±0.009 | 0.809±0.006 | 0.834±0.004 | **0.853±0.003** |
| | OE | 0.638±0.023 | 0.579±0.006 | 0.581±0.008 | 0.537±0.014 | 0.573±0.015 | 0.579±0.006 | **0.539±0.007** |
| | Cov | 0.323±0.015 | 0.273±0.002 | 0.241±0.003 | 0.232±0.009 | 0.269±0.006 | 0.237±0.005 | **0.215±0.009** |
| ESPGame | AP | 0.244±0.005 | 0.246±0.002 | 0.289±0.003 | 0.294±0.004 | 0.297±0.002 | 0.294±0.004 | **0.312±0.004** |
| | 1-RL | 0.808±0.002 | 0.818±0.002 | 0.832±0.001 | 0.832±0.002 | 0.832±0.001 | 0.828±0.002 | **0.847±0.002** |
| | AUC | 0.813±0.002 | 0.824±0.002 | 0.836±0.001 | 0.835±0.002 | 0.836±0.001 | 0.833±0.002 | **0.852±0.002** |
| | OE | 0.657±0.013 | 0.661±0.003 | 0.604±0.005 | 0.567±0.008 | 0.561±0.007 | 0.566±0.009 | **0.549±0.010** |
| | Cov | 0.452±0.004 | 0.429±0.003 | 0.426±0.004 | 0.394±0.004 | 0.407±0.003 | 0.410±0.004 | **0.372±0.005** |
| IAPRTC12 | AP | 0.237±0.003 | 0.261±0.001 | 0.305±0.004 | 0.325±0.004 | 0.323±0.001 | 0.317±0.003 | **0.339±0.003** |
| | 1-RL | 0.833±0.002 | 0.848±0.001 | 0.862±0.002 | 0.873±0.004 | 0.873±0.001 | 0.870±0.001 | **0.886±0.002** |
| | AUC | 0.835±0.001 | 0.850±0.001 | 0.864±0.002 | 0.875±0.004 | 0.874±0.000 | 0.872±0.001 | **0.888±0.004** |
| | OE | 0.648±0.008 | 0.610±0.005 | 0.568±0.008 | 0.543±0.008 | 0.532±0.002 | 0.557±0.005 | **0.537±0.002** |
| | Cov | 0.436±0.005 | 0.408±0.004 | 0.403±0.004 | 0.335±0.007 | 0.351±0.001 | 0.352±0.003 | **0.312±0.007** |
| Mirflickr | AP | 0.490±0.012 | 0.551±0.002 | 0.570±0.002 | 0.612±0.005 | 0.589±0.005 | 0.594±0.005 | **0.611±0.002** |
| | 1-RL | 0.803±0.008 | 0.844±0.001 | 0.856±0.001 | 0.871±0.002 | 0.865±0.003 | 0.863±0.004 | **0.875±0.001** |
| | AUC | 0.787±0.012 | 0.837±0.001 | 0.846±0.001 | 0.856±0.003 | 0.853±0.003 | 0.849±0.004 | **0.862±0.001** |
| | OE | 0.489±0.022 | 0.415±0.003 | 0.369±0.004 | 0.331±0.007 | 0.358±0.008 | 0.363±0.007 | **0.338±0.006** |
| | Cov | 0.428±0.013 | 0.369±0.002 | 0.360±0.001 | 0.323±0.003 | 0.333±0.003 | 0.348±0.007 | **0.319±0.003** |

necessity of considering possible missing views and labels during the model design process.

To demonstrate the effectiveness of our uncertainty-aware pseudo-labeling strategy, similar to [17], we empirically analyze our strategy in different way on various datasets, each with half of the views and labels missing, and figure 5 illustrates our findings. Fig.5a shows the relationship of mean absolute error (MAE) and the uncertainty of output predictions on three different datasets. It is evident that with the increase of uncertainty, the MAE of outputs increases accordingly, thereby confirming our model's ability to accurately estimate prediction uncertainty. Fig.5b, Fig.5c and Fig.5d explore the MAE, accuracy, and the number of pseudo-labels selected across different epochs of training. It is noteworthy that the accuracy of the pseudo-labels chosen by our model remains consistently high, irrespective of the number of labels selected during the training process, which proves the effectiveness of our self-paced-based selection strategy and the robustness of our uncertainty-aware pseudo-labeling strategy. Furthermore, Fig.5c and Fig.5d show that our pseudo-labeling strategy can select more credible pseudo-labels

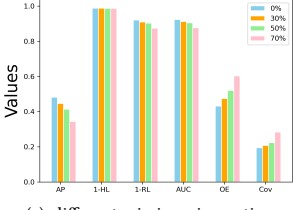 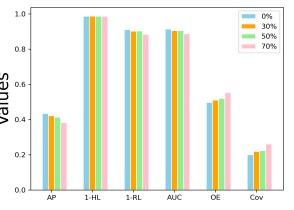

(a) different missing-view ratios      (b) different missing-label ratios

Figure 3: Experimental results on the Corel5k dataset: (a) different missing-view ratios and a 50% missing-label ratio and (b) different missing-label ratios and a 50% missing-view ratio.

compared to conventional pseudo-labeling strategy. This observation illustrates the superiority of our uncertainty-aware pseudo-labeling strategy in identifying and selecting the most credible pseudo-labels for our model, further reinforcing its effectiveness.

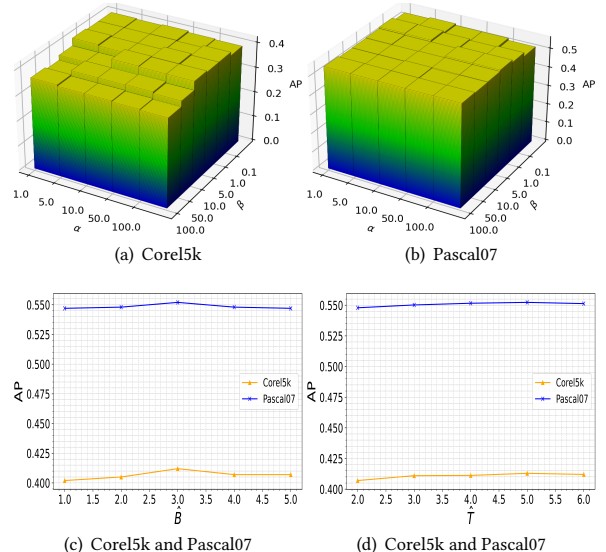

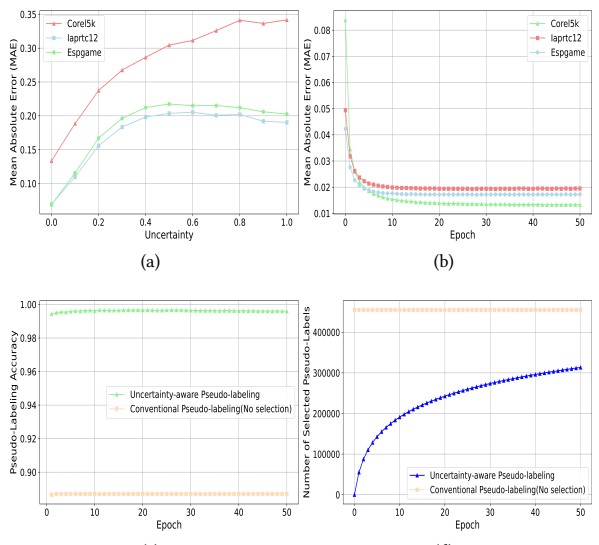

Figure 4: The AP values for hyper-parameters $\alpha$ and $\beta$ on the Corel5k (Fig. a) and Pascal07 (Fig. b) datasets; AP values for hyper-parameter $\hat{B}$ on both Corel5k and Pascal07 datasets (Fig. c); and AP values for hyper-parameter $\hat{T}$ on Corel5k and Pascal07 (Fig. d) datasets are presented. Both datasets contain 50% available views and labels, with a 70% training sample rate.

Figure 5: Fig. a shows the relationship between output uncertainty and MAE values on various datasets; Fig. b shows the MAE values of pseudo-labels during the training stage on different datasets. Fig. c and Fig.d compare the accuracy and number of pseudo-labels selected by conventional pseudo-labeling strategy and our uncertainty-aware pseudo-labeling strategy on the Corel5k dataset.

## 4.3 Hyper-parameter Study

**On $\alpha$ and $\beta$**: Our model's overall loss involves two hyperparameters, $\alpha$ and $\beta$. We explored their sensitivity by varying their values and reporting the corresponding AP values on the Corel5k and Pascal07 datasets, with conditions of 50% available instances per view, 50% missing labels, and 70% training samples. As shown in Fig 4a and Fig 4b, the optimal ranges for $\alpha$ and $\beta$ on Corel5k are [1,10] and [0.1,5], and on Pascal07, they are [1,10] and [0.1,10], respectively.

**On $\hat{B}$ and $\hat{T}$**: From Fig 4c, it can be seen that too large or too small of the number of average view tokens is not conducive to achieving optimal performance. Based on experiments, we set $\hat{B}$ as 3 for all datasets. Fig 4d shows the AP value versus hyper-parameters $\hat{T}$, which can be seen that when $\hat{T}$ is greater than 4, our method is not sensitive to it, so we set $\hat{T} = 5$ for all five datasets.

## 4.4 Ablation Study

The ablation experiments are conducted on the Corel5k and ESPGame datasets, with 50% missing views, 50% missing labels, and 70% training samples. We sequentially removed AVT, $\ell_{gcs}$, $\ell_{gcl}$, and the uncertainty-aware pseudo-labeling strategy (PLS). Notably, we leverage pseudo-labels selected from PSL as extra supervised information to calculate the classification loss in our network. Consequently, we remove the cross-entropy loss between the predictions and pseudo-labels to validate the effectiveness of our PSL. The results presented in Table 2, reflect two key observations: (i) Each component of our UPDGD-Net is crucial and contributes positively, emphasizing the significance of every component in enhancing

multi-label classification performance. (ii) The most significant enhancement is observed with the integration of the SPL.

## 5 Conclusion

In this paper, we propose an uncertainty-aware pseudo-labeling and dual graph driven network (UPDGD-Net) for iMvMLC tasks, designing an uncertainty-aware pseudo-labeling strategy to harness the semantic information in missing labels. We also utilize multi-label topological relationships to guide representation learning in embedding space and propose the average view token to enhance the multi-view information synthesis. Extensive experiments validate the effectiveness of our approach.

Table 2: The ablation experiment result on Corel5k dataset and Pascal07 dataset.

| Backbone | AVT | $L_{gcs}$ | $L_{gcl}$ | PLS | Core15k | | Pascal07 | |
|---|---|---|---|---|---|---|---|---|
| | | | | | AP | AUC | AP | AUC |
| ✓ | | | | | 0.366 | 0.890 | 0.520 | 0.835 |
| ✓ | ✓ | | | | 0.378 | 0.891 | 0.528 | 0.839 |
| ✓ | | ✓ | | | 0.388 | 0.892 | 0.524 | 0.837 |
| ✓ | | | ✓ | | 0.384 | 0.893 | 0.524 | 0.837 |
| ✓ | | | | ✓ | 0.380 | 0.903 | 0.534 | 0.844 |
| ✓ | ✓ | ✓ | | | 0.394 | 0.895 | 0.530 | 0.838 |
| ✓ | | ✓ | ✓ | | 0.396 | 0.899 | 0.533 | 0.842 |
| ✓ | ✓ | ✓ | ✓ | | 0.404 | 0.899 | 0.538 | 0.843 |
| ✓ | ✓ | ✓ | ✓ | ✓ | **0.413** | **0.905** | **0.552** | **0.853** |

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

Received 20 February 2007; revised 12 March 2009; accepted 5 June 2009