# OpenReview forum: "Uncertainty-Aware Pseudo-Labeling and Dual Graph Driven Network for Incomplete Multi-View Multi-Label Classification"
_acmmm.org/ACMMM/2024/Conference — MM2024 Poster_

### Official Review · Reviewer_iPRm · 2024-05-14

**Rating:** 5
**Confidence:** 3

**Summary:**

The paper proposes a novel approach for handling incomplete multi-view multi-label classification (iMvMLC) tasks. This is particularly relevant in fields such as medical imaging and bioinformatics, where missing data are common due to various uncertainties in data collection and labeling. Key novelties include Uncertainty-Aware Pseudo-Labeling (uses a strategy that employs uncertainty estimation to generate pseudo-labels for missing labels, enhancing the learning process despite incomplete data)  and Dual Graph Constraints (leverages a label matrix to apply constraints on feature embeddings at both the view and label levels, helping maintain the intrinsic structure of the data during feature extraction). The model uses transformer encoders to learn multi-view representations and label interactions. It outperforms 6 previous methods on 5 benchmark datasets in experiments. Ablation studies validate the importance of the graph constraints and pseudo-labeling components.

**Strengths:**

(1) The paper introduces an innovative approach to handling incomplete multi-view and multi-label data through uncertainty-aware pseudo-labeling and dual graph constraints. This method is novel as it leverages the rich semantic information in the label space to preserve the intrinsic data structure during representation learning, and it augments the incomplete label information by selecting credible pseudo-labels based on prediction uncertainty.
(2) The paper provides comprehensive experimental results to validate the effectiveness of the proposed methods. Testing the model on five different datasets and comparing it with several SOTA methods, demonstrates its practical utility and robustness.
(3) One of the key strengths of the paper is its focus on robustness to incomplete data, which is a common issue in practical applications. The methodology developed is particularly relevant for areas like medical imaging and bioinformatics, where missing data are frequent.

**Limitations:**

(1) The proposed UPDGD-Net has a complex architecture involving multiple transformer encoders, graph constraints, and pseudo-labeling. The high complexity makes it difficult to interpret and analyze the model's behavior and the contributions of individual components. Iterative uncertainty estimation via Monte Carlo dropout sampling and the transformer architectures can be computationally expensive, especially for large datasets. The paper does not discuss the model's efficiency and costs.
(2) I feel like there are many hyperparameters in these proposed methods. Although some hyperparameters are discussed in Section 4.3, others remain undiscussed. For example, the pseudo-labeling strategy involves multiple hyperparameters (e.g., temperature coefficient, growth rate, uncertainty threshold) whose selection criteria and sensitivity analysis could be better explained.
(3) The results of Figure 3 could be more thoroughly explained. Since there are many evaluation metrics, the authors could first state which one is the primary metric. Then, they could provide more details about the figure. For example, they could explain why the OE and Cov metrics increase as the missing-view ratio and missing-label ratio continue to grow.

**Suitability:**

3

---

### Official Review · Reviewer_87Qu · 2024-05-19

**Rating:** 2
**Confidence:** 3

**Summary:**

This paper introduces UPDGD-Net, a framework designed to handle multi-view multi-label learning with imcomplete view/label. This network features an uncertainty-aware pseudo-labeling strategy to fill missing labels with pseudo-labels, achieved by Monte-Carlo Dropout to quantify uncertainty. It also imposes dual graph constraints on embedded features at both the view and label levels to maintain the inherent data structure. Additionally, transformer-based modules are used for cross-view aggregation and multi-label classification, utilizing average view tokens (AVT) to ensure consistency. Experiments on five datasets demonstrate that UPDGD-Net outperforms existing methods, effectively addressing incomplete data issues and improving classification accuracy.

**Strengths:**

1. This paper studies an interesting yet understudied area, which is multi-view multi-label learning with incomplete views and labels.

2. This paper is overall well-written and well-structured.

3. The proposed modules undergo ablation studies and suggest the effectiveness of the individual modules.

**Limitations:**

Major issues:

1. This paper proposed a lot of different modules - yet the proposition of those modules is not well-motivated. I do not see strong associations between those modules and the problem of missing view/label.

2. Some relevant works [1,2] are omitted - is there a reason why those works are not included or compared? More importantly, the problem of both missing view and missing label has been studied by [2], but there is no discussion on the limitations and differences between the proposed method and [2] whatsoever.

3. The proposed uncertainty-aware pseudo-labeling straightforwardly applies the Mote Carlo dropout, which does not appear to be novel, in my opinion.

Minor issues:

1. The notations and problem definition should not be included in related works.

[1] Multi-View Multi-Label Learning with View-Specific Information Extraction, IJCAI 2019.

[2] Deep Double Incomplete Multi-View Multi-Label Learning With Incomplete Labels and Missing Views, TNNLS 2023.

**Suitability:**

2

---

### Official Review · Reviewer_dmKs · 2024-05-21

**Rating:** 4
**Confidence:** 2

**Summary:**

This paper considers multi-view multi-label classification (MvMLC) problem with missing labels and views of data and proposes a graph-based framework UPDGD-Net. The proposed UPDGD-Net models and aligns data samples by constructing a cross-view and a cross-label graph, as well as a pseudo-labeling strategy to enhance model performance. Comprehensive experiments illustrate the performance of the proposed method.

**Strengths:**

1. The paper is well written and easy to follow.
2. A comprehensive architecture that models both samle and label correlation with a contrastive graph learning framework.
3. Experimental results demonstrate the superiority of the UPDGD-Net framework.

**Limitations:**

1. As the authors have frankly mentioned, most of the designs in UPDGD-Net are originated from previous works on multi-view learning [1], multi-label learning [2], as well as other label-utlization techniques [3], which raise the concerns over the novelty of this work.
2. The motivation of using a Monte-Carlo Dropout strategy, as well as how the proposed UPDGD-Net would address MvMLC tasks with incomplete information require further explanations.
3. The performance of baseline models with complete label and view information should also be concluded to make a comprehensive comparison.




[1] Incomplete MultiView Multi-Label Learning via Label-Guided Masked View- and Category-Aware Transformers, AAAI 2023

[2] General Multi-label Image Classification with Transformer, CVPR 2021

[3] Collaborative Learning of Label Semantics and Deep Label-Specific Features for Multi-Label Classification, TPAMI 2022

**Suitability:**

3

---

### Official Review · Reviewer_sxyh · 2024-05-24

**Rating:** 3
**Confidence:** 4

**Summary:**

The paper introduces the UPDGD-Net framework named of uncertainty-aware pseudo-labeling and dual graph driven network, which focuses on a practical problem for incomplete multi-view learning. It tackles two main challenges: How to obtain an effective representation with incomplete views and How to choose credible pseudo-labels for missing labels. Compared with the existing methods, the authors design dual graph constraints on the embedded features of both view-level and label-level. The experimental results on several real-world datasets demonstrate the effectiveness of the proposed model.

**Strengths:**

1. The paper is well-motivated since missing is a common and significant problem in real-world multi-view learning. The model design for the missing problem in this paper is very reasonable, offering a highly comprehensive coverage.

2. The paper provides a well-structured overview, making it easy to follow for readers. Besides, the implementation details of the selected technique are described in detail.

3. The experiments were conducted in several benchmark datasets, showing the superiority of the proposed method.

**Limitations:**

1. About novelty: Beyond the framework innovation, there seems to be absence of original techniques and methods, with the approach primarily consisting of an amalgamation of existing techniques. Besides, the runnable code is not provided, making readers hard for reproduction of Label-Guided Dual Constraints and Pseudo-Labeling module.

2. About clarity: Lack analysis about how previous researches addressed the problem about tackling Pseudo-Labeling for missing labels. Besides, why uncertainty-based pseudo-labeling can be seen as the credible pseudo-labels? How to define credible pseudo-labels? The authors should give theoretical proof or discuss and analyze.

3. About soundness: Lack of time analysis and how is the complexity of this method compared to the comparison method? It seems that the proposed algorithm has a high time complexity.

4. About experiments: From my point of view, the setting of the experiments is unfair and biased. Since the proposed UPDGD-Net compared with the comparison methods has both feature and label completion modules, the experimental settings should be conducted under different missing rates, like [1]. Why only consider the case that the missing rate is 50%? How about the performance under missing rate from 60% to 90%. It seems that the effectiveness of the model can be better reflected in the case of extreme missing (90% missing rate).

[1] Lin, Zhenghong, et al. "Contrastive Intra-and Inter-Modality Generation for Enhancing Incomplete Multimedia Recommendation." Proceedings of the 31st ACM International Conference on Multimedia. 2023.

**Suitability:**

2

---

### Meta-Review · Area_Chair_RnvB · 2024-06-29

**Recommendation:** Accept (Poster)
**Confidence:** 4

**Metareview:**

This paper received wa, wa, ba, br from reviewers after rebuttal. Reviewers believe that this paper is the first work to learn discriminative topological information under incomplete scenarios. There are also some concerns from reviewers such as motivation and related works. Authors should solve these issues in their final version.